# Body Composition in Women after Radical Mastectomy

**DOI:** 10.3390/ijerph17238991

**Published:** 2020-12-02

**Authors:** Jacek Wilczyński, Piotr Sobolewski, Rafał Zieliński, Magdalena Kabała

**Affiliations:** Laboratory of Posturology, Collegium Medicum, Jan Kochanowski University, 25-369 Kielce, Poland; piotrsobolewski@poczta.onet.pl (P.S.); rafal.zielinski@ujk.edu.pl (R.Z.); magdalena.kabala@onet.eu (M.K.)

**Keywords:** radical mastectomy, body composition, electrical bioimpedance

## Abstract

The aim of the study was to analyse the body composition among women after radical mastectomy. The body compositions of 30 women after radical mastectomy (study group) were compared with those of 30 healthy females (control group). The method of electrical bioimpedance was used to analyse body composition. The significant differences between the groups, unfavourable for women, following mastectomy concerned body mass (*p* = 0.021), BMI (*p* = 0.049), fat mass (%) (*p* = 0.007), fat mass (kg) (*p* = 0.005), total body water (%) (*p* = 0.002), left upper limb fat mass (*p* = 0.013) as well as right upper limb fat mass (*p* = 0.022). The body composition of women after radical mastectomy was significantly worse compared to the control group. The majority of subjects were overweight and had high levels of body fat. Abnormal body composition is a modifiable risk factor of breast cancer; therefore, improving lifestyle is important in the prevention and treatment of this disease. There is a need for education, dietary supervision and physical activity in women following radical mastectomy. The innovation of our study was the use of the modern bioelectrical impedance analysis (BIA) method, which does not cause ionisation and is a gold standard in the field of body composition analysis. In future research, we plan to broaden the assessment of lifestyle and the significance of diet and physical activity in the prevention and treatment of breast cancer.

## 1. Introduction

Breast cancer is the most common malignant neoplasm diagnosed in women. Approximately 1 million new cases are diagnosed in the world each year. About 500,000 new and existing patients will die from breast cancer [1,2]. Postmenopausal women with abnormal body composition and obesity are at a higher risk of developing this disease [3,4]. In obese women, additional diagnostic difficulties occur, as well as an increased risk of complications during treatment and a greater likelihood of recurrence of the disease. There is also a greater risk of developing cancer of the other breast [5,6]. The risk of breast cancer in obese postmenopausal women is three times higher than that in lean women with normal body composition [7,8]. Adipose tissue is an endocrine structure that secretes biologically active substances. The association between body composition disorders and breast cancer is due to the metabolic interaction between oestrogens from peripheral aromatisation, the role of the insulin/insulin-like growth factor axis and the function of adipocytes as an endocrine organ. In many studies, a correlation has been demonstrated between excess body mass and breast cancer. Overweightness and obesity are characterized as excessive accumulation of fatty tissue in the body [9].

Obesity, defined by WHO as a chronic disease, is a long-term inflammatory process [10]. As a result, excessive secretion of pro-inflammatory substances (TNF-α, IL-6, visfatin, resistin) occurs, while the concentration of anti-inflammatory substances (adiponectin, IL-1, IL-10) is reduced [11]. This leads to increased cell proliferation, inhibition of apoptosis, stimulation of angiogenesis and damage to genetic material. All of this increases the risk of developing breast cancer. People with excess body mass often eat inconsistently with the generally accepted health recommendations [12]. A high-fat, low-fibre diet with a high glycaemic index and a low consumption of vegetables and fruits promotes the development of both obesity and breast cancer. The problem of excessive adipose tissue concerns not only anti-cancer prevention, but it also affects the course of treatment and prognosis [13]. Consequently, there is a need for nutritional education among cancer patients and dietary supervision in the case of weight reduction for obese cancer patients [14,15].

The limited efficiency of clinical trials and a reluctance to participate in screening tests result in a higher stage of the disease at diagnosis, which translates into worse treatment outcomes [16]. Obese people are at a greater risk of any type of anaesthesia, an increased risk of technical difficulties during breast cancer surgery, in wound infection and in the rate of wound revision due to bleeding, as well as an increased risk in the number of thromboembolic complications and the risk of limb oedema following lymphatic surgery. The course of the disease is significantly worse in obese women, which is confirmed by the results of clinical trials [11]. Abnormal body composition is a modifiable risk factor for breast cancer. Improving lifestyle to reduce overweightness and obesity may have significant impact on reducing the incidence of and deaths from breast cancer [17,18].

Currently, the significance of regular physical activity in the prevention of breast cancer is more and more emphasized. The aim of systematic physical activity is to support the rehabilitation process [19]. It is about, among others, restoring normal muscular-ligamentous-joint functions. Properly selecting the intensity of exercise is an important element of primary and secondary prevention of breast cancer [20]. In many publications, the significant, positive effect of physical activity in reducing the incidence of breast cancer is described. It is indicated that the occurrence of breast cancer is greatly lower in physically active women [21]. Physical activity also prevents post-mastectomy complications [22]. Moreover, it positively influences the limitation and reduction of lymphoedema, prevents overweightness and improves physical fitness as well as body statics [23].

The aim of the study was to analyse the body composition of women following radical mastectomy.

## 2. Materials and Methods

The study comprised 60 women aged 45 to 60. The study group (1) consisted of 30 women following radical mastectomy (right, left), whereas in the control group (2), there were 30 healthy women with similar anthropometric parameters and of similar age. All examined patients were right-handed. The inclusion criterion for the study was 3 to 4 years following the surgery. In the study group, 13 (43.33%) women underwent left radical mastectomy, while 17 (56.67%) women underwent right radical mastectomy. The adjuvant treatment included radiotherapy in 23 (76.67%) women, followed by hormone therapy in 22 (73.33%) women and chemotherapy in 19 (63.33%) women. The patients did not have lymphedema.

Basic somatic features were examined. Body height was measured with an anthropometer to the nearest 5 mm, while body mass was estimated with an electronic scale to the nearest 0.5 kg. Based on the obtained data, body mass index was calculated. The method of electrical bioimpedance (BIA) was used to analyse body composition. BIA (bioelectrical impedance analysis) was used to measure impedance, which is the electrical resistance comprising the resistance and reactance of the tissues through which a low electric current (≤1 mA) is passed. The phenomenon of resistance is related to the specific resistance of individual tissues, while reactance is mainly caused by the electric capacity of cell membranes, which, due to their structure, act as capacitors. Adipose tissue has a low water content and, therefore, is more resistant to an electrical signal, while muscle tissue has a high water content and, thus, conducts electricity with less resistance. However, factors such as ambient temperature, previous exercise, fluid intake, health state, menstrual cycle, medications, alcohol, caffeine, etc., may also affect tissue hydration levels and test results. The time of day can also influence water content as well as fluid distribution in the body. This is why measurements are most consistent when taken before the evening meal. In order to obtain the most accurate measurements, they were always carried out at the same time by the same trained person (between 18:00 and 20:00 p.m.) and under the same conditions, e.g., always before a meal. The BIA method used in TANITA analysers is a gold standard in the field of body composition analysis. The accuracy of the measurement is as follows: body mass 50 g; fatty tissue content 0.1%; muscle tissue content 50 g; body water content 0.1%. We also bore in mind a few contraindications, e.g., pregnancy and pacemakers [24].

The test was performed using the TANITA MC-780 device. As a result of the measurement, the following variables were obtained: body mass (kg), metabolic age, body mass index (BMI), fat mass FM (%), fat mass—FM (kg), fat-free mass—FFM (kg), muscle mass—MM (kg), total body water—TBW (kg) and total body water—TBW (%), muscle mass of the left upper limb—MMLUL (kg), muscle mass of the right upper limb—MMRUL (kg), left upper limb fat mass—LULFM (kg) and right upper limb fat mass—RULFM (kg). The study was performed in 2019 at the Posturology Laboratory. All research procedures were performed in accordance with the 1964 Declaration of Helsinki and with the consent of the University Bioethics Committee No. 31/2018. Measurement data were collected on an MS Excel spreadsheet. After pre-treatment, they were imported to the Statistica 13 program. The analysis included anthropometric and body composition variables, both in the study (1) and control group (2). The selection of statistical analysis methods was determined by the type of analysed variables (quantitative variables). For the analysed quantitative variables, both describing the age of the subjects, body mass and height, metabolic age and body composition, arithmetic means (x), standard deviations (s), medians (Me) and extreme values were calculated. Normality of distribution for the variables was checked using the Kolmogorov–Smirnov test. The non-parametric Mann–Whitney U test was used to demonstrate the differences between the study group (1) and the control group (2). This test allows a comparison between each of the observations against the median, not the mean. Therefore, when reporting its results, attention was paid to the median value in both groups, and conclusions were drawn on this basis. The level of statistical significance was *p* < 0.05.

## 3. Results

In the study group (1), there were 17 (56.67%) women post right-sided mastectomy and 13 (43.33%) women post left-sided mastectomy. The mean age in the study group (1) was 55.07 years, and in the control group (2) 50.27 years. There were no significant differences in body height between the study (1) and control (2) group (*p* = 0.622). The mean body mass for the study group (1) was 73.01 kg, and for the control group (2) it was 65.95 kg. Body mass differed significantly between the groups (*p* = 0.021).

BMI in the test group (1) was 27.56 kg/m^2^, and in the control group (2) it was 24.96 kg/m^2^. There were significant differences in BMI between the study group (1) and the control group (2) (*p* = 0.049), as well as significant differences in metabolic age between the studied groups (*p* = 0.001). BMI in the test group (1) was 27.56 kg/m^2^, and in the control group (2), 24.96 kg/m^2^. There were significant differences in BMI between the study (1) and control (2) group (*p* = 0.049) (Table 1). 

The mean fat mass (%) in the study group (1) was 34.03%, and in the control group (2), 29.16%. There was a significant difference in fat mass (%) between the two groups (*p* = 0.007). The mean fat mass (kg) in the study group (1) was 24.99 kg, and in the control group (2) it was 19.33 kg. There was a significant difference in fat mass (kg) between groups (*p* = 0.005). The mean lean mass (kg) for the study group (1) was 48.59 kg, and for the control group (2), 46.96 kg. There were no significant differences in lean mass (kg) between groups (*p* = 0.224).

The mean muscle mass (kg) for the study group (1) was 47 kg, and for the control group (2), this totalled 43.97 kg. There were no significant differences in muscle mass (kg) between groups (*p* = 0.109). The mean total water content (kg) in the study group (1) was 34.42 kg, and in the control group (2) it was, 33.37 kg. There were no significant differences in total water content (kg) between the groups (*p* = 0.236).

The mean total water content (%) in the study group (1) was 47.59%, while in the control group (2), this equalled 51.16%. There was a significant difference in total water content (%) between groups (*p* = 0.002). Metabolic age in the study group (1) was 51.00 years, and in the control group it was 39.83 years (2). There were significant differences in metabolic age between the examined groups (*p* = 0.001) (Table 1).

Left upper limb muscle mass (kg) in the study group (1) was 2.27 kg, and in the control group (2) this amounted to 2.18 kg. There were no significant differences in left upper limb muscle mass (kg) between the groups (*p* = 0.254). The muscle mass of the right upper limb (kg) in the study group (1) was 2.29 kg, and in the control group (2), 2.18 kg. There were no significant differences in right upper limb muscle mass (kg) between the groups (*p* = 0.213). The mass of the left upper limb fat (kg) in the test group (1) was 1.54 kg, and in the control group it was 1.09 kg (2). There was a significant difference in left upper limb fat mass (kg) between groups (*p* = 0.013). The mass of the right upper limb fat (kg) in the study group (1) was 1.42 kg, while in the control group (2) it was 1.03. There was a significant difference in the mass of right upper limb fat (kg) between the examined groups (*p* = 0.022) (Table 1).

## 4. Discussion

One of the significant risk factors for developing breast cancer, apart from genetic determinants, is inappropriate diet and the associated accumulation of adipose tissue [25]. Excess body fat is associated with an overgrowth of adipocytes. In women with a normal BMI, the overgrowth of breast adipocytes correlates with inflammation of white adipose tissue, increased levels of aromatase (an enzyme that limits the rate of oestrogen biosynthesis) and increased blood leptin levels [26,27]. Importantly, it is suggested that insulin resistance, inflammation of the adipose tissue of the breast, elevated aromatase expression and elevated leptin levels play roles in the pathogenesis of obesity-related breast cancer [28].

Obesity increases the risk of breast cancer, especially in postmenopausal women [29]. The menopausal period is associated with hormonal changes that directly affect a woman’s body composition. First of all, the share of adipose tissue in the total body mass and that located in the abdominal area increases [30]. In the case of most women, following a mastectomy, the distribution of adipose tissue occurs in the android form where fat is mainly located in the upper body, especially in the abdomen and chest [31]. The results of our study, as well as previous reports, confirm that after a mastectomy a significant percentage of women are overweight or obese [32]. Among obese women, deficits in maintaining balance may result from various functional determinants of postural stability [33]. This may be due to a slower response because of increased inertia of body segments, increased joint stiffness or decreased mobility due to excess body fat [34].

In the research by McTiernan et al. [35], it has been shown that obesity and a sedentary lifestyle increase the risk of breast cancer by about 25–33%. Additionally, the results of a meta-analysis by Chan et al. [36] demonstrated that obese women were at a 35% higher risk of dying from breast cancer compared to those with normal body mass. Treatment of breast cancer is associated with an increase in body fat as well as a decrease in lean body mass and bone mineral density. These changes can increase the risk of brittle fractures and the formation of osteoporosis, and they may lead to the reoccurrence of cancer. Maintaining proper norms of body composition should be part of important preventive measures among women, especially those over 50 [36].

In our study, taking into account bioelectrical impedance analysis (BIA) in women after unilateral mastectomy, a greater amount of fat mass was observed when compared to the control group. Statistical significance was demonstrated in terms of fat mass (kg) (*p* = 0.005) and fat mass (%) (*p* = 0.007) between the study and control groups. Additionally, the content of fat mass in the left and right upper limbs among women after radical mastectomy was higher in the study than in the control group. Statistical significance was demonstrated between the groups *p* = 0.013 and *p* = 0.022, respectively. In our research, as well as in that by Coin et al. [37], it was confirmed that the right arm has slightly less fat tissue than the left arm.

In a study by Gomes et al. [38], 95 women were examined (including 49 women from the study group, six months after surgery for breast cancer and 46 women from the control group). In that study, women who underwent left-sided radical mastectomy showed a greater total lean mass compared to women who underwent right-sided radical mastectomy. In women with excessive lymphedema, an increase in the volume of adipose and lean tissue on the affected side was observed. Among patients after breast cancer treatment, there was often a combination of weight loss with a build-up of fat, i.e., so-called sarcopenic obesity. The full aetiology of the loss of lean body mass requires further research; however, it may be associated with worse metabolic outcomes associated with the development of cardiovascular diseases or metabolic syndrome [38].

Vance et al. [39], based on an analysis of the literature from 1975 to 2009, showed that weight gain is more common in women receiving adjuvant chemotherapy, especially among those undergoing longer treatment, and it seems to be particularly pronounced in premenopausal women. With or without weight gain, there are adverse changes in body composition, including increased fat and loss of lean tissue [39].

Additionally, van den Berg et al. [40] performed a meta-analysis in which they also indicated that there is significant weight gain during chemotherapy in women with breast cancer. Moreover, chemotherapy promotes disorders of muscle metabolism (deregulation of adenosine triphosphate, cytokines and depletion of satellite cells occurs) and also leads to the destruction of muscle, which may weaken muscle strength and reduce the level of fitness [40].

Although physical activity is a lifestyle element crucial in maintaining physical fitness and directly influences the perceived quality of life, reports on this subject differ [41]. In one study, no significant differences were found in the level of physical activity undertaken before or after mastectomy. This treatment did not affect the frequency, type or form of rest among the respondents. The participants presented a general, average level of knowledge about the influence of physical activity on the reduction in disease recurrence risk. Only half of the respondents were aware of the influence of exercise on general fitness and physical health as well as mental health [42]. In a different study, it has been shown that women change their eating habits when diagnosed with a disease but, at the same time, do not increase physical activity despite its beneficial effects on quality of life and prevention of cancer recurrence [43]. In yet another study, we found information indicating that mastectomy surgery and the lack of habitual movement in this group of people cause a decrease in physical activity, and thus, a lack of it being regularly performed [44]. According to the respondents, the main obstacles hindering physical activity are household chores, lack of time or poor health. Other authors also cite limitations such as fatigue, pain and reluctance to exercise. There are also barriers closely related to the effects of the disease and its treatment, including the following: secondary lymphoedema, fear of pain, lack of information about permitted types of activity, bad mood, depression and apathy. However, there are also studies in which it is shown that having had a past mastectomy significantly increased the frequency of undertaking physical exercise. This may indicate an increase in awareness of the need to perform physical activity [44].

Despite the proven influence of the increased content of adipose tissue in the body as a significant factor in the aetiology of breast cancer, the distribution of this tissue has not been fully elucidated and requires further research [45,46]. Assessing women’s body composition can be an important indicator of metabolic disorders and disease progression.

## 5. Conclusions

The body composition of women after radical mastectomy was significantly worse compared to the control group. Most of the subjects were overweight and had high levels of body fat. Abnormal body composition is a modifiable risk factor for breast cancer; therefore, improving lifestyle is important in the prevention and treatment of this disease. There is a need for education, dietary supervision and physical activity among women after having a radical mastectomy. The innovation of our study was the use of the modern BIA method, which does not cause ionisation and is a gold standard in the field of body composition analysis. In future studies, we plan to broaden the assessment of lifestyle and the importance of diet and physical activity in the prevention and treatment of breast cancer.

## Figures and Tables

**Table 1 ijerph-17-08991-t001:** Somatic features and body composition in the study (1) and control (2) groups.

Variables	Descriptive Statistics of the Analysed Scales						Mann–Whitney U Test
	Group	X	SD	Min	Med	Max	
Age (years)	1	55.07	4.71	45.00	55.50	60.00	*p =* 0.001
	2	50.27	5.13	45.00	49.00	60.00	
Height (cm)	1	163.03	4.49	152.00	164.00	173.00	*p* = 0.622
	2	162.40	4.76	152.00	164.00	176.00	
Body mass (kg)	1	73.01	12.93	46.50	69.75	104.50	*p* = 0.021
	2	65.95	11.21	49.90	62.30	92.90	
Metabolic age	1	51.00	13.04	29.00	50.00	83.00	*p* = 0.001
	2	39.83	10.73	27.00	34.00	63.00	
BMI (kg/m^2^)	1	27.56	5.32	18.20	26.65	40.10	*p* = 0.049
	2	24.96	3.73	18.60	24.40	32.80	
Fat mass (%)	1	34.03	8.96	13.50	33.55	66.50	*p* = 0.007
	2	29.16	8.58	14.00	27.65	60.20	
Fat mass (kg)	1	24.99	8.39	6.30	23.65	44.00	*p* = 0.005
	2	19.33	7.11	7.00	18.05	36.50	
Fat-free mass (kg)	1	48.59	5.57	40.20	47.80	60.50	*p* = 0.224
	2	46.96	5.38	39.70	45.70	62.40	
Muscle mass (kg)	1	47.00	6.78	38.10	46.30	69.60	*p* = 0.109
	2	43.97	6.29	24.50	43.40	59.30	
Total body water (kg)	1	34.42	3.97	28.30	33.85	43.00	*p* = 0.236
	2	33.37	3.85	28.00	32.45	44.40	
Total body water (%)	1	47.59	4.68	39.50	47.20	61.30	*p* = 0.002
	2	51.16	4.50	43.30	51.60	61.30	
Muscle mass of left upper limb	1	2.27	0.31	1.80	2.20	2.80	*p* = 0.254
	2	2.18	0.30	1.80	2.15	3.00	
Muscle mass of right upper limb	1	2.29	0.34	1.80	2.20	3.00	*p* = 0.213
	2	2.18	0.30	1.80	2.10	3.10	
Fat mass of left upper limb	1	1.54	0.77	0.20	1.30	3.50	*p* = 0.013
	2	1.09	0.52	0.30	1.00	2.20	
Fat mass of upper right limb	1	1.42	0.69	0.20	1.20	3.30	*p* = 0.022
	2	1.03	0.48	0.20	0.90	2.00	

## Data Availability

The data used to support the findings of this study are available from the corresponding author upon request.

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
