# Peer review of "Body Composition in Women after Radical Mastectomy"

_ijerph, 2020, doi:10.3390/ijerph17238991_

Round 1
Reviewer 1 Report
The Manuscript “Body composition in women after radical mastectomy” is pertinent and well-structured despite the novelty of this study should be clarified, in the revised version.
The following extensive revision is needed before the publication, as well as a general editing of English language and style
Methods:
According to several studies (i.e. Kroker et al. 2011), the methodology to perform the electrical bioimpedance is very precise, could the authors specify their precaution to avoid errors during the measurements.
Line 70 translate the word antropometerto
Could be interesting to include information about the level of physical activity and/or the dietary habits of the patients that can highly affect the results reported.
Statistical Analysis:
There is no explanation of the statistical analysis approach used, please provide.
Results:
Why the author decided to report the same results both in table and in graphic form? can be disorienting
Conclusion:
It’s seem a repetition of the discussion. In this section the author should highlight the innovation of the study and explain the limitations.
Author Response
Thank you very much for your insightful comments and recommendations. We are very grateful for the time you have devoted to the review. We have implemented the required changes in the text. Below, please find a point-by-point response to the comments. Once more, thank you for the insight which, we are sure, has significantly improved the substantive value of our manuscript.
Point 1. The Manuscript “Body composition in women after radical mastectomy” is pertinent and well-structured despite the novelty of this study should be clarified, in the revised version.
Response 1: We have implemented the required changes in the text.
Point 2. The following extensive revision is needed before the publication, as well as a general editing of English language and style
Response 2: We have done extensive proofreading as well as general editing of the English language and style.
However, if the text is still not linguistically correct, we will ask for a correction from the editorial office of the Journal.
Point 3. Methods: According to several studies (i.e. Kroker et al. 2011), the methodology to perform the electrical bioimpedance is very precise, could the authors specify their precaution to avoid errors during the measurements.
Response 3: Body composition analysis using the BIA (bioelectrical impedance analysis) method consisted in measuring the impedance, which is the electrical resistance composed of the resistance and reactance of the tissues through which a low electric current (≤1 mA) is passed. The phenomenon of resistance is related to the specific resistance of individual tissues, while reactance is mainly due to the electric capacity of cell membranes, which, because of their structure, act as capacitors. Adipose tissue has a low water content and thus, is more resistant to an electrical signal, while muscle tissue has a high water content and therefore, conducts electricity with less resistance. However, factors such as ambient temperature, previous exercise, fluid intake, health, menstrual cycle, medications, alcohol, caffeine, etc., can also affect tissue hydration and test results. The time of day may also influence the water content as well as the fluid distribution in the body. This is why measurements are most consistent when taken before an evening meal.
In order to obtain the most accurate measurements, they were always performed at the same time by the same trained person (between 6:00 and 8:00 p.m.). and under the same conditions, e.g. always before a meal. The BIA method used in TANITA analysers is a gold standard in the field of body composition analysis. The accuracy of the measurement is as follows: body mass 50 g; fatty tissue content 0.1%; muscle tissue content 50 g; body water content 0.1%. We also bore in mind a few contraindications, e.g. pregnancy, pacemakers.
Point 4. Line 70 translate the word antropometerto.
Response 4: I corrected this sentence: Basic somatic features were examined. Body height was measured with an anthropometer to the nearest 5 mm, while body mass was estimated with an electronic scale to the nearest 0.5 kg.
Point 5. Could be interesting to include information about the level of physical activity and/or the dietary habits of the patients that can highly affect the results reported.
Response 5: Before the onset of the disease, most of the respondents did not know that nutrition plays such an important role in the prevention and development of breast cancer. In general, the subjects did not know that the diet should be properly balanced, preferably free of animal fats. It should be based on fresh vegetables, fruits, wholesome proteins, fiber and compounds with antioxidant and carcinogenic properties, e.g. vitamin C, A, E, selenium, lycopene and polyphenols. However, they declared that after the mastectomy procedure, they tried to follow the nutritional medical recommendations.
We supplemented the article with the issue of the importance of exercise, which can modify the lifestyle and body composition of women, especially as it is indeed a key part of our research. We have included works on women's physical activity and body composition as well as breast cancer and physical activity in the ‘Introduction’ and ‘Discussion’ sections.
Introduction: Currently, the importance of undertaking regular physical activity in the prevention of breast cancer is more and more emphasized. The aim of systematic physical activity is to support the rehabilitation process (Kruk, J. Intensity of recreational physical activity in different life periods in relation to breast cancer among women in the region of Western Pomerania. Contemporary Oncology 2012, 16 (6), 576-581). This is related, among others, to restoring normal muscular-ligamentous-joint functions. A properly selected intensity of efforts is a significant element of primary and secondary breast cancer prevention (Ligibel, JA; Partridge, A.; Giobbie-Hurder, A.; Golshan, M.; Emmons, K.; Winer, EP. Physical activity behaviors in women with newly diagnosed ductal carcinoma-in-situ in a longitudinal cohort study. Ann Surg Oncol 2009, 16, 106-112). In numerous publications, a significant positive effect of physical activity in reducing the incidence of breast cancer is described. Breast cancer occurs significantly less frequently in physically active women (Hsieh, C.; Sprod, LK; Hydock, DS et al. Effects of a Supervised Exercise Intervention on Recovery from Treatment Regimens in Breast Cancer Survivors. Oncol Nurs Forum 2008, 35 ( 6), 909-915). Physical activity also prevents complications after mastectomy (Pierce, J.; Stefanick, M.; Flatt, S. Greater survival after breast cancer in physically active women with high vegetable-fruit intake regardless of obesity. J Clin Oncol 2007, 17, 234551). In addition, it has a positive effect on the limitation and reduction of lymphoedema, it prevents overweightness, improves physical fitness and body statics (Penttinen, HM; Saarto, T.; Kellokumpu-Lehtinen, P.; Blomqvist, C.; Huovinen, R.; Kautiainen, H. et al. Quality of life and physical performance and activity of breast cancer patients after adjuvant treatments. Psychooncology 2011, 20 (11), 1211-20).Discussion: Although physical activity is an element of lifestyle crucial in maintaining physical fitness and directly influencing the perceived quality of life, reports on this subject differ (Prokopowicz, K.; KozdroÅ„, E.; Prokopowicz, G.; Molik, B.; Berk, A.; Mucha, J. Conditions of physical activity undertaken by women after surgical breast cancer treatment. Hygeia Public Health 2018, 53 (1), 100-105). In one study, no significant differences were found in the level of physical activity undertaken before or after mastectomy. This treatment did not affect the frequency, type or form of rest among the respondents. The participants presented a general, average level of knowledge about the influence of physical activity on the reduction of the risk of disease recurrence. Only half of the respondents were aware of the influence of exercise on general fitness and physical as well as mental health (12. Ridan, T.; Zdebska, S .; Ogrodzka, K.; Opuchlik, A. Evaluation of physical activity level in women after single breast mastectomy. Problems of Hygiene and Epidemiology 2015, 96 (1), 181-186). In a different study, it has been shown that women change their eating habits when diagnosed with disease, but at the same time, do not increase physical activity despite its beneficial effects on quality of life and prevention of cancer recurrence (13. Hashemi Bani, SH; Karimi, S.; Mahboobi, H. Lifestyle changes for prevention of breast cancer. Electron Physician 2014, 3 (3), 894-905). In yet another study, we find information indicating that mastectomy surgery and the lack of the habit of needing to move in this group of people cause a decrease in physical activity, and thus, the lack of its regular performace (Irwin, ML; McTiernan, A.; Manson, JE et al. Physical activity and survival in postmenopausal women with breast cancer: results from the women’s health initiative. Cancer Prev Res 2011,4 (4), 522-529). According to the respondents, the main obstacles hindering physical activity are household chores, lack of time or poor health. Other authors also cite limitations such as fatigue, pain and reluctance to exercise. There are also barriers closely related to the effects of the disease and its treatment: secondary lymphoedema, fear of pain, lack of information about permitted types of activity, bad mood, depression and apathy. However, there are also studies in which it is shown that past mastectomy significantly increased the frequency of undertaking physical exercise. This may indicate an increase in awareness of the need to perform physical activity (Prokopowicz, K.; KozdroÅ„, E.; Prokopowicz, G.; Molik, B.; Berk, A.; Mucha, J. Conditions of physical activity performed by women after surgical breast cancer treatment. Hygeia Public Health 2018, 53 (1), 100-105).
Point 6. Statistical Analysis:There is no explanation of the statistical analysis approach used, please provide.
Response 6: The measurement data were collected using an MS Excel spreadsheet. After pre-treatment, they were imported to the Statistica 13 program. The analysis included anthropometric and body composition variables, both in the study (1) and the control group (2). The selection of statistical analysis methods was determined by the type of the analysed variables (quantitative variables). For the analysed quantitative variables, both describing the age of the subjects, body mass and height, metabolic age and body composition, arithmetic means (x), standard deviations (s), medians (Me) and extreme values, ​were calculated. These variables were tested for normality of distribution via the Kolmogorov-Smirnov test. The non-parametric Mann-Whitney U test was used to demonstrate the differences between the test group (1) and the control group (2). This test compares each of the observations against the median, not the mean. Therefore, when reporting its results, attention was paid to the median value in both groups and conclusions were drawn on this basis. The level of statistical significance was p<0.05.
Point 6. Results:Why the author decided to report the same results both in table and in graphic form? can be disorienting.
Response 6: As suggested by the Reviewer, in order not to confuse the readers, we have removed the graphic presentation of the results.
Point 7. Conclusion:It’s seem a repetition of the discussion. In this section the author should highlight the innovation of the study and explain the limitations.
Response 7: As suggested by the Reviewer, the conclusions have been corrected so that they are not a repetition of the ‘Discussion’. We tried to highlight the innovativeness of our research and explain its limitations
The body composition of women after radical mastectomy was significantly worse compared to the control group. Most of the subjects were overweight and had high levels of body fat. Abnormal body composition is a modifiable risk factor for breast cancer, therefore, improving lifestyle is important in the prevention and treatment of this disease. There is a need for education, dietary supervision and physical activity among women after radical mastectomy. The innovation of our study was the use of the modern BIA method, which does not cause ionisation and is a gold standard in the field of body composition analysis. In future studies, we plan to broaden the assessment of lifestyle and the importance of diet and physical activity in the prevention and treatment of breast cancer.
Once more, we are exceptionally grateful for your in-depth review of our article. Your insight and comments will definitely allow for an increase in the substantive value of the manuscript. Thank you for your devoted time and effort.
Yours sincerely,
Assoc. Prof. UJK Jacek Wilczyński, Ph.D.

Reviewer 2 Report
The article entitled “Body Composition in Women after Radical Mastectomy” is very interesting because of the novelty of the subject. In it, the authors compare two samples of women to check the differences that exist between a group of mastectomized women and another group of healthy women.
Next, I will present my review of your article, with the intention of making it better.
INTRODUCTION
The introduction is good. It talks about how obesity is a risk factor for the appearance of cancer and its treatment. It collects a series of manuscripts that provide a good basis to the subject, however, when they talk about lifestyle and the variables that influence it (nutritional education)but they do not mention physical exercise as a variable that can modify lifestyle and even body composition (a key part of the study). In fact, they only mention physical activity in one quote in the discussion.
We recommend the inclusion of some references on:
- physical activity, women and body composition
- breast cancer and physical activity/rowing/etc.
METHOD.
It would be interesting if between the control group and the study group there were not so much difference between the age and weight of the participants (perhaps it would be convenient to reduce the number of participants and make these parameters similar).
There is a difference of 7 kg between the groups, that will cause significant differences in the BMI and it is possible that in other variables. We recommend, as far as possible, to try to homogenize the samples and to include tests of normality of both samples.
It would be interesting to exclude in the contrast between groups (K-S) the variable: age. Since it usually gives significant differences.
On the other hand, we recommend the use of DXA instead of electrical Bioimpedance, since the former gives much more precise results.
RESULTS
Well thought out, but the graphics are repetitive. The information already appears in table 1. It would be interesting to eliminate the figures because of their reiterative content and low resolution.
If possible, it would be interesting to make a more in-depth study of the results
DISCUSSION
Acceptable. The effect of physical activity on the lifestyle and body composition of women with and without breast cancer should be explored
CONCLUSIONS
They are scarce, and the practical application of their study is lacking. Please work on this aspect of the study. Your study and the discovery of body composition changes in women with breast cancer is very interesting.
Finally I would like to congratulate you for the work, and I hope to see it published soon.
Author Response
Thank you very much for your insightful comments and recommendations. We are very grateful for the time you have devoted to the review. We have implemented the required changes in the text. Below, please find a point-by-point response to the comments. Once more, thank you for the insight which, we are sure, has significantly improved the substantive value of our manuscript.
Point 1. The article entitled “Body Composition in Women after Radical Mastectomy” is very interesting because of the novelty of the subject. In it, the authors compare two samples of women to check the differences that exist between a group of mastectomized women and another group of healthy women.
Next, I will present my review of your article, with the intention of making it better.
Response 1: We have implemented the required changes in the text.
Point 2. Introduction: The introduction is good. It talks about how obesity is a risk factor for the appearance of cancer and its treatment. It collects a series of manuscripts that provide a good basis to the subject, however, when they talk about lifestyle and the variables that influence it (nutritional education)but they do not mention physical exercise as a variable that can modify lifestyle and even body composition (a key part of the study). In fact, they only mention physical activity in one quote in the discussion. We recommend the inclusion of some references on: physical activity, women and body composition; breast cancer and physical activity/rowing/etc.
Response 2: We have added the issue of exercise as a variable that may modify the lifestyle and even body composition of women, all the more so as it is a key part of our research. We have included papers on women's physical activity and body composition, as well as breast cancer and physical activity in the ‘Discussion’ section.
Point 3. METHOD.It would be interesting if between the control group and the study group there were not so much difference between the age and weight of the participants (perhaps it would be convenient to reduce the number of participants and make these parameters similar).There is a difference of 7 kg between the groups, that will cause significant differences in the BMI and it is possible that in other variables. We recommend, as far as possible, to try to homogenize the samples and to include tests of normality of both samples.It would be interesting to exclude in the contrast between groups (K-S) the variable: age. Since it usually gives significant differences.On the other hand, we recommend the use of DXA instead of electrical Bioimpedance, since the former gives much more precise results.
Response 3: In our future-planned study, we will exclude the variable of age from the contrast between groups (C-S). We also agree that the difference in body mass between groups is too great (7 kg), which may cause significant differences in BMI and other body composition variables. We will take this into account when planning further research.
In our trial, we attempted to show that the risk and development of breast cancer is significantly higher in overweight women. Abnormal body composition is a modifiable risk factor of breast cancer. Improving lifestyle to reduce overweightness and obesity may have significant impact on reducing the incidence of breast cancer and deaths caused by it [18,19]. Women after radical mastectomy are generally overweight. When recruiting participants for testing, we did not follow their body mass but the course of mastectomy.
The respondents asked about the invasiveness of this study (does it not cause ionisation?). The women paid special attention to the fact that our test methods were non-invasive. They asked if Tanita ionises and how it works. This is one of the reasons why we chose the BIA method.BIA (bioelectrical impedance analysis) measured the impedance, which is the electrical resistance composed of the resistance and reactance of the tissues through which a low electric current (≤1 mA) is passed. The phenomenon of resistance is related to the specific resistance of individual tissues, while reactance is mainly caused by the electric capacity of cell membranes, which, due to their structure, act as capacitors. Adipose tissue has a low water content and therefore, is more resistant to an electrical signal, while muscle tissue has a high water content and thus, conducts electricity with less resistance. However, factors such as ambient temperature, previous exercise, fluid intake, health state, menstrual cycle, medications, alcohol, caffeine, etc., may also affect tissue hydration levels and test results. The time of day can also influence water content as well as fluid distribution in the body. This is why measurements are most consistent when taken before the evening meal. In order to obtain the most accurate measurements, they were always carried out at the same time by the same trained person (between 6:00 and 8:00 p.m.) and under the same conditions, e.g. always before a meal. The BIA method used in TANITA analysers is the gold standard in the field of body composition analysis. The accuracy of the measurement is as follows: body mass: 50 g; fatty tissue content 0.1%; muscle tissue content 50 g; body water content 0.1%. We also bore in mind a few contraindications, e.g. pregnancy, pacemakers.
Point 4. RESULTSWell thought out, but the graphics are repetitive. The information already appears in table 1. It would be interesting to eliminate the figures because of their reiterative content and low resolution.If possible, it would be interesting to make a more in-depth study of the results.
Response 4: As suggested by the Reviewer, in order not to duplicate the results, their graphic presentation has been removed. We have described the applied statistical methods in more detail. The measurement data were collected using an MS Excel spreadsheet. After pre-treatment, they were imported to the Statistica 13 program. The analysis included anthropometric and body composition variables, both in the study (1) and the control group (2). The selection of statistical analysis methods was determined by the type of the analysed variables (quantitative variables). For the analysed quantitative variables, both describing the age of the subjects, body mass and height, metabolic age and body composition, arithmetic means (x), standard deviations (s), medians (Me) and extreme values, ​were calculated. These variables were tested for normality of distribution via the Kolmogorov-Smirnov test. The non-parametric Mann-Whitney U test was used to demonstrate the differences between the test group (1) and the control group (2). This test compares each of the observations against the median, not the mean. Therefore, when reporting its results, attention was paid to the median value in both groups and conclusions were drawn on this basis. The level of statistical significance was p<0.05.
Point 5. DISCUSSIONAcceptable. The effect of physical activity on the lifestyle and body composition of women with and without breast cancer should be explored.
Response 5: We supplemented the article with the issue of the significance of exercise, which can modify the lifestyle and body composition of women, especially as it is indeed a key part of our research. We have included works on women's physical activity and body composition as well as breast cancer and physical activity in the ‘Introduction’ and ‘Discussion’ sections. Introduction: Currently, the importance of undertaking regular physical activity in the prevention of breast cancer is more and more emphasized. The aim of systematic physical activity is to support the rehabilitation process (Kruk, J. Intensity of recreational physical activity in different life periods in relation to breast cancer among women in the region of Western Pomerania. Contemporary Oncology 2012, 16 (6), 576-581). This is related, among others, to restoring normal muscular-ligamentous-joint functions. A properly selected intensity of efforts is a significant element of primary and secondary breast cancer prevention (Ligibel, JA; Partridge, A.; Giobbie-Hurder, A.; Golshan, M.; Emmons, K.; Winer, EP. Physical activity behaviors in women with newly diagnosed ductal carcinoma-in-situ in a longitudinal cohort study. Ann Surg Oncol 2009, 16, 106-112). In numerous publications, a significant positive effect of physical activity in reducing the incidence of breast cancer is described. Breast cancer occurs significantly less frequently in physically active women (Hsieh, C.; Sprod, LK; Hydock, DS et al. Effects of a Supervised Exercise Intervention on Recovery from Treatment Regimens in Breast Cancer Survivors. Oncol Nurs Forum 2008, 35 ( 6), 909-915). Physical activity also prevents complications after mastectomy (Pierce, J.; Stefanick, M.; Flatt, S. Greater survival after breast cancer in physically active women with high vegetable-fruit intake regardless of obesity. J Clin Oncol 2007, 17, 234551). In addition, it has a positive effect on the limitation and reduction of lymphoedema, it prevents overweightness, improves physical fitness and body statics (Penttinen, HM; Saarto, T.; Kellokumpu-Lehtinen, P.; Blomqvist, C.; Huovinen, R.; Kautiainen, H. et al. Quality of life and physical performance and activity of breast cancer patients after adjuvant treatments. Psychooncology 2011, 20 (11), 1211-20). Discussion: Although physical activity is an element of lifestyle crucial in maintaining physical fitness and directly influencing the perceived quality of life, reports on this subject differ (Prokopowicz, K.; KozdroÅ„, E.; Prokopowicz, G.; Molik, B.; Berk, A.; Mucha, J. Conditions of physical activity undertaken by women after surgical breast cancer treatment. Hygeia Public Health 2018, 53 (1), 100-105). In one study, no significant differences were found in the level of physical activity undertaken before or after mastectomy. This treatment did not affect the frequency, type or form of rest among the respondents. The participants presented a general, average level of knowledge about the influence of physical activity on the reduction of the risk of disease recurrence. Only half of the respondents were aware of the influence of exercise on general fitness and physical as well as mental health (12. Ridan, T.; Zdebska, S .; Ogrodzka, K.; Opuchlik, A. Evaluation of physical activity level in women after single breast mastectomy. Problems of Hygiene and Epidemiology 2015, 96 (1), 181-186). In a different study, it has been shown that women change their eating habits when diagnosed with disease, but at the same time, do not increase physical activity despite its beneficial effects on quality of life and prevention of cancer recurrence (13. Hashemi Bani, SH; Karimi, S.; Mahboobi, H. Lifestyle changes for prevention of breast cancer. Electron Physician 2014, 3 (3), 894-905). In yet another study, we find information indicating that mastectomy surgery and the lack of the habit of needing to move in this group of people cause a decrease in physical activity, and thus, the lack of its regular performace (Irwin, ML; McTiernan, A.; Manson, JE et al. Physical activity and survival in postmenopausal women with breast cancer: results from the women’s health initiative. Cancer Prev Res 2011,4 (4), 522-529). According to the respondents, the main obstacles hindering physical activity are household chores, lack of time or poor health. Other authors also cite limitations such as fatigue, pain and reluctance to exercise. There are also barriers closely related to the effects of the disease and its treatment: secondary lymphoedema, fear of pain, lack of information about permitted types of activity, bad mood, depression and apathy. However, there are also studies in which it is shown that past mastectomy significantly increased the frequency of undertaking physical exercise. This may indicate an increase in awareness of the need to perform physical activity (Prokopowicz, K.; KozdroÅ„, E.; Prokopowicz, G.; Molik, B.; Berk, A.; Mucha, J. Conditions of physical activity performed by women after surgical breast cancer treatment. Hygeia Public Health 2018, 53 (1), 100-105).
Point 6. Statistical Analysis: There is no explanation of the statistical analysis approach used, please provide.
Response 6: Measurement data were collected on an MS Excel spreadsheet. After pre-treatment, they were imported to the Statistica 13 program. The analysis included anthropometric and body composition variables, both in the study (1) and control group (2). The selection of statistical analysis methods was determined by the type of analysed variables (quantitative variables). For the analysed quantitative variables, both describing the age of the subjects, body mass and height, metabolic age and body composition, arithmetic means (x), standard deviations (s), medians (Me) and extreme values ​​were calculated. Normality of distribution for the variables was checked using the Kolmogorov-Smirnov test. The non-parametric Mann-Whitney U test was used to demonstrate differences between the study group (1) and the control group (2). This test allows to compare each of the observations against the median, not the mean. Therefore, when reporting its results, attention was paid to the median value in both groups and conclusions were drawn on this basis. The level of statistical significance was p<0.05.
Point 6. CONCLUSIONS They are scarce, and the practical application of their study is lacking. Please work on this aspect of the study. Your study and the discovery of body composition changes in women with breast cancer is very interesting. Finally I would like to congratulate you for the work, and I hope to see it published soon.
Response 6: As suggested by the Reviewer, the ‘Conclusions’ section has been corrected. The body composition of women after radical mastectomy was significantly worse compared to the control group. Most of the subjects were overweight and had high levels of body fat. Abnormal body composition is a modifiable risk factor for breast cancer, therefore, improving lifestyle is important in the prevention and treatment of this disease. There is a need for education, dietary supervision and physical activity among women after radical mastectomy. The innovation of our study was the use of the modern BIA method, which does not cause ionisation and is a gold standard in the field of body composition analysis. In future studies, we plan to broaden the assessment of lifestyle and the importance of diet and physical activity in the prevention and treatment of breast cancer.
Once more, we are exceptionally grateful for your in-depth review of our article. Your insight and comments will definitely allow for an increase in the substantive value of the manuscript. Thank you for your devoted time and effort.
Yours sincerely,
Assoc. Prof. UJK Jacek Wilczyński, Ph.D.

Round 2
Reviewer 1 Report
All concerns have been sufficiently addressed by the authors. I recommend publication of the manuscript.